# Study on the Penetration Power of ZrO_2_ Toughened Al_2_O_3_ Ceramic Composite Projectile into Ceramic Composite Armor

**DOI:** 10.3390/ma15082909

**Published:** 2022-04-15

**Authors:** Rui Yang, Kuiwu Li, Likui Yin, Kai Ren, Yu Cheng, Taotao Li, Jianping Fu, Taiyong Zhao, Zhigang Chen, Jinlong Yang

**Affiliations:** 1College of Mechatronic Engineering, North University of China, Taiyuan 030051, China; yrmiffy@163.com (R.Y.); cy909677@163.com (Y.C.); 2Institute for Civ-Mil Integration & Collaborative Innovation, North University of China, Taiyuan 030051, China; lkwmxw@126.com; 3National Defense Key Discipline Laboratory of Underground Target Damage Technology, North University of China, Taiyuan 030051, China; 18335178769@139.com (L.Y.); zs_991109@163.com (T.Z.); tj85tj@163.com (Z.C.); 4School of Environment and Safety Engineering, North University of China, Taiyuan 030051, China; ltt1127k@163.com; 5The State Key Laboratory of New Ceramics and Fine Processing, Tsinghua University, Beijing 100084, China; jlyang@mail.tsinghua.edu.cn

**Keywords:** penetration ability, ZrO_2_ toughened Al_2_O_3_, composite projectile, ceramic composite armor

## Abstract

This work aims to improve the penetration ability of a 14.5 mm standard armor-piercing projectile into ceramic/armor steel (Al_2_O_3_/RHA) composite armor. To this end, ZrO_2_ toughened Al_2_O_3_(ZTA) is prepared as the material for bullet tips, utilizing in situ solidification injection molding that is realized via ceramic dispersant hydrolytic degradation. The penetration power of ZTA ceramic composite projectile, compared with standard armor, against 15 mm armor steel (RHA) and 30 mm Al_2_O_3_/RHA composite armor, is studied by ballistics testing combined with numerical simulation. The Tate theory is optimized and then employed to calculate the penetration depth and bullet core’s residual mass when ZTA ceramic composite projectile penetrates into Al_2_O_3_/RHA composite armor. The results show that when penetrating RHA of 15 mm, the penetration area of ZTA ceramic composite projectile into RHA increases by 27.59% and the exit area by 42.93%. While the standard projectile fails to penetrate the 30 mm Al_2_O_3_/RHA composite armor, the ZTA ceramic composite armor-piercing projectile succeeds, with the mass loss reduced by 66.67% over the standard one. The ZTA ceramic composite bullet has a better performance than the standard bullet in penetrating RHA and Al_2_O_3_/RHA composite armors. The test results, simulation, and theoretical analysis are consistent. This study has practical values for engineering applications to design new ceramic composite bullets.

## 1. Introduction

As one of the most effective protective structures nowadays, Al_2_O_3_/RHA composite armor can defend against high-speed fragments and small-medium caliber bullets. Therefore, it has been widely introduced in protecting personnel, armored vehicles, fighters, and other military targets [1,2,3,4,5]. Due to the high strength and rigidity of the ceramic panel, the locking effect and interface defeat phenomenon occur on the contact surface of the projectile target at the initial stage of penetration [6,7,8,9], deforming or even breaking up conventional projectiles laterally, and thus consuming part of their kinetic energy. After the ceramic is broken into a ceramic cone, the projectile’s energy is dispersed and acts over a wider area. At the same time, the erosion of the projectile body and consumption of its mass result in a large loss of specific kinetic energy of the projectile and a sharp reduction of penetration ability [10,11,12,13]. As ceramic composite armor technology develops and is widely applied, traditional armor-piercing weapons face new challenges regarding their penetration power. Therefore, it is urgent to find new armor-piercing materials and match new armor-piercing structures.

ZTA ceramic material has excellent mechanical properties such as hardness, impact resistance, wear resistance, etc. [14,15,16], which underpins its use as an armor-piercing projectile tip material. For example, Kai et al. [17] used the 5.8 mm standard projectile tip, where Ø 2 mm and Ø 5 mm ZrO_2_ ceramic balls were applied, to penetrate the ceramic/Kevlar composite target plate at 750 m/s. The composite projectile improves the penetration ability of the ceramic composite target but fails to penetrate through the plate. Khan et al. [18] studied the ballistic performance of a 4340 steel projectile when it hit a 99.5% Al_2_O_3_ ceramic plate at the speed of 52 m/s~275 m/s and also the crack growth and target resistance of the ceramic target plate. Jinfeng et al. [19] determined ballistic response using a 12.7 mm standard projectile to penetrate the light ceramic-metal composite armor through experiments and numerical simulation.

At present, studies on ceramic materials have mainly focused on the application fields of ballistic performance and armor protection [20,21,22,23,24,25,26], but few of them pay attention to the application of ceramic materials in armor-piercing projectile tips to improve the penetration ability. In this study, ZTA ceramic material is matched with the standard armor-piercing projectile core of 14.5 mm to prepare a ZTA composite armor-piercing projectile, while the projectile shape and mass are maintained to be consistent. The Al_2_O_3_ ceramic panel of the Al_2_O_3_/RHA composite armor is attacked by a ZTA ceramic bullet tip. Then, the RHA backplate is penetrated by the metal projectile core so that highly-efficient damage is caused to the armored target, and its protection ability is weakened. The research results are of high value in theory and engineering applications concerning the design of the armor-piercing projectile tip and penetrating ceramic composite armor.

## 2. Test Preparation

### 2.1. Selection of ZTA Bullet Tip Material

Al_2_O_3_ ceramic material has been introduced into many fields thanks to its excellence in hardness, mass, wear and chemical corrosion resistance, high-temperature stability, low material cost, etc. However, limited by its brittleness, regular Al_2_O_3_ ceramic materials are often severely damaged and fractured or even broken under an impact less than the Hugoniot Elastic Limit (HEL), with the flexural strength and fracture toughness registering at only about 380 MPa and 3.5 M·m^1/2^, respectively. On the other hand, the toughed ZTA ceramics meet the requirements of ceramic parts under the high-speed penetration, boasting 558 MPa of the flexural strength, 5.72 MPa·m^1/2^ of the fracture toughness, and 13.8 GPa of the hardness as well as good wear resistance.

ZTA ceramic is a kind of multi-phase fine ceramic material prepared by introducing a certain amount of phase change material (PCM) ZrO_2_ into the parent phase matrix-Al_2_O_3_, which not only highlights the characteristics of high toughness and strength of ZrO_2_ ceramics but also retains the high hardness of Al_2_O_3_ ceramics. Meanwhile, its comprehensive mechanical properties, especially wear resistance, are significantly enhanced. Based on the study of ZTA ceramics with fine grains prepared from industrial-grade raw materials, ZTA wear-resistant ceramic parts suitable for high-speed impact environments have been successfully developed through pressure casting with new dispersants. Those parts characterize the high green body density, excellent sintering performance, as well as uniform and dense microstructure. Thanks primarily to its reasonable composition and uniform and dense structure, ZTA ceramics show good processing properties, thus laying the foundation for its use as a projectile warhead material.

The mechanical properties of ZTA ceramics are influenced by different sintering temperatures and ZrO_2_ contents. The best mechanical properties can be achieved at the sintering temperature of 1550 °C, with flexural strength and fracture toughness of ZTA ceramics registering 708 MPa and 5.8 MPa·m^1/2,^ respectively, representing an increase of about 117% and 49% compared to the Al_2_O_3_ matrix. As the content of ZrO_2_ increases, ZTA ceramics can gradually get better fracture strength and toughness. When the content of ZrO_2_ reaches 20%, this material is fully densified, in which case, significant progress can be achieved in fine crystallization, dispersion, and phase transformation toughening effects of ZrO_2_. This means that this content of ZrO_2_ is the optimum for ZTA ceramics to have the best mechanical properties, with the relative density, three-point bending strength, and fracture toughness reaching the maximum. The specific change trend of the mechanical properties of ZTA with ZrO_2_ content is shown in Figure 1. The influence of ZrO_2_ content on ZTA fracture is shown in Figure 2.

### 2.2. Preparation Process of ZTA Projectile Material

The raw material used was an α-A1_2_O_3_ powder with a purity of 98.8%; the particle size and *d*_50_ of which were 0.07 μm~5.86 μm and 0.88 μm, respectively. The purity of the ZrO_2_ powder was 94.6%, with a particle size of 0.04 μm~4.02 μm and *d*_50_ of 0.43 μm. The organic monomer was acrylamide; the cross-linking agent was N,N′-methylene-bis-acrylamide; the initiator was ammonium sulfate; the catalyst was N,N,N′,N′-tetramethylethylenediamine; and the dispersant was ammonium citrate C_6_H_5_O_7_(NH_4_)_3_. All the above reagents were analytically pure.

The ZTA ceramic projectile was prepared with in situ solidification injection molding through ceramic dispersant hydrolytic degradation, the process flow of which is shown in Figure 3. A standby premix solution was prepared by mixing 14.5 wt% monomers, 0.5 wt% cross-linking agents, and deionized water. ZrO_2_ and A1_2_O_3_ powder made by the liquid phase method were mixed with a mass fraction of 1:4. Next, a certain amount of ceramic powder, dispersant, and monomer premix was mixed, and ball milled for 24 h to obtain a ceramic slurry with a solid content of 44%. Then, the slurry was poured out, with catalyst and initiator being added successively and stirred evenly before being injected into the mold to be cured and molded to obtain a ceramic blank, which was machined to a given size. Finally, the green body was heated to 900 °C at a heating rate of 0.5 °C/min for binder removal and then heated to 1550 °C at the heating rate of 1 °C/min for sintering to obtain the ceramic projectile with the required size and weight. The microstructure of the prepared green body and ZTA ceramics is shown in Figure 4. In Figure 4a, it can be seen that the microstructure of the green body was uniform without large particles and defects caused by agglomeration. Figure 4b shows that the microstructure of the polished surface of the ZTA ceramic was uniform and dense. The physical and mechanical properties of the ZTA ceramics are shown in Table 1.

### 2.3. Micro Fracture Analysis of ZTA Material

Brittle fracture is the main failure mode of ceramics, in which the failure process is crack propagation. Polycrystalline materials are susceptible to many defects such as dislocations and pores, while the dislocation intersections and pores form stress concentration points, ultimately resulting in cracks. Meanwhile, the differences in thermal expansion of the grains with different orientations in polycrystalline can also lead to cracks. The thermodynamic barrier between crack nucleation and propagation needs to be compensated for by the energy from thermal fluctuation, but the energy takes time to accumulate. Due to the limited stress action time during dynamic loading, the crack nucleation and propagation cannot be completed as under static conditions, meaning that the energy released is unable to achieve material fracture, and therefore, the stress must be increased [26]. The micro appearance of fracture of Al_2_O_3_ ceramics (a, b) and ZTA ceramics (c, d) after high-speed impact is shown in Figure 5. In Figure 5a,b, the main fracture form of Al_2_O_3_ ceramics is an intergranular fracture, which occurs on the crystal surface. The fracture is in the shape of “crystal sugar” without signs of plastic deformation. It can be seen from Figure 5c,d that the main fracture form of ZTA ceramics is a transgranular ductile fracture, which consumes more energy. In sum, ZTA ceramic has better impact resistance as a projectile material.

### 2.4. Experiment of Penetration Power

The penetration experiments of the 14.5 mm armor-piercing projectile on an RHA (armor steel) target and on Al_2_O_3_/RHA composite armor were carried out. The structures of the two projectiles are shown in Figure 6, with the one on the left showing the standard armor-piercing projectile and the right showing the ZTA ceramic composite projectile. These two projectiles are identical in terms of overall dimensions and quality. The ZTA composite projectile is composed of a ceramic projectile, metal core, lead can, and jacket, with the core material being T12A. The structural parameters of the projectile are shown in Table 2.

Two typical target plates selected in the experiment are: rolled homogeneous armor steel (RHA) (Size: 200 mm × 200 mm × 15 mm) and Al_2_O_3_/RHA composite armor. According to the structural diagram shown in Figure 7, from the bullet front surface to the back surface, there are 1.5 mm glass fiber layer, 15 mm Al_2_O_3_ ceramic plate, 1.5 mm glass fiber layer, and 15 mm armor steel. The glass fiber layer serves as the buffer layer to attenuate stress wave, anti-caving, and flame retardant. The ceramic plate, generally a barrier layer, mainly acts as ceramic material in resisting high pressure and high hardness, causing the bullet to suffer from mass abrasion, fracture, or deflection, ultimately reducing its subsequent kinetic energy. The steel backplate is mainly used to strengthen the structural strength and improve the anti-penetration performance. The ceramic plate is bonded by the glass fiber layer, Al_2_O_3_ ceramic block, and glass fiber layer with the adhesive of epoxy/polyamide resin. The dimensions of the ceramic plate and armored steel backplate are 200 mm × 200 mm × 15 mm.

A 14.5 mm smooth-bore ballistic gun was used in the experiment, and the laser velocimeter independently developed by the North University of China was used as the speed measuring device. The diagram of the experiment setup is shown in Figure 8.

## 3. Simulation Calculation

### 3.1. Simulation Model

In the comparative analysis of the abilities of ZTA ceramic composite projectiles and standard projectiles penetrating the target plate, a finite element model was established by TrueGrid and calculated by AUTODYN. A Smoothed Particle Hydrodynamics (SPH) algorithm was used for the ZTA ceramic warhead and Al_2_O_3_ ceramic target plate; the Lagrange algorithm was used for the bullet core and RHA target plate; and pressure-outflow boundary conditions were applied on the model boundary. The projectile target simulation model is shown in Figure 9.

### 3.2. Material Parameters

In the simulation calculation, the JH-2 material model was adopted in the ZTA ceramic projectile and Al_2_O_3_ ceramic target plate, and the Johnson–Cook material model in the core and RHA material. The Gruneisen equation of state was employed to describe the dynamic response behavior, and the Puff state equation and von Mises strength model of the glass fiber. The parameters of ZTA ceramic materials are shown in Table 3, and other parameters are shown in reference [22]. The parameters of the RHA, metal core, and jacket are shown in references [27,28].

## 4. Results and Analysis

### 4.1. Penetration of RHA Armor Steel

#### 4.1.1. Experiment Results

A ballistic gun test was designed to investigate the penetration ability of the ZTA ceramic composite projectile and standard projectile into armor steel. The two armor-piercing projectiles with different structures penetrated 15 mm homogeneous armor-steel targets at the speed of 1000 ± 15 m/s. The test was repeated three times, and the average result was taken as the standard for evaluating the projectile power. The test data are shown in Table 4, and the comparison of the typical damage effects of the armor steel target is shown in Figure 10.

It can be seen from Table 4 and Figure 10 that, that in all six rounds, the penetrations into 15 mmRHA were successful. However, compared to the standard projectile, the average hole entrance area of the ZTA composite projectile penetrating RHA increased by 27.59%, and the average hole exit area by 42.93%. This is because when the ZTA ceramic composite projectile penetrates into homogeneous armor steel, the warhead ceramic is broken under the high pressure generated by high-speed penetration at the moment of contact between the projectile and the target. Part of the target plate materials is taken away when splashing in the opposite direction under the action of the kinetic energy of the projectile body. At the same time, the broken ceramics abrase and erode the surface of the target plate. All these would contribute to a larger entry hole of the target plate. In the recovered cores of the ZTA composite armor-piercing projectile and standard armor-piercing projectile, their respective average residual mass was 25.9 g and 20.5 g, accounting for 72.35% and 50.87% of the initial core mass, respectively. The average mass loss of the ZTA composite projectile was 43.72% less than that of the standard projectile. The average residual velocities of the two projectiles were 516.7 m/s and 434.1 m/s, respectively, taking up 51.67% and 43.41% of the initial velocity. The average residual velocity and residual kinetic energy of the ZTA composite projectile were 19.03% and 78.99% higher than that of the standard projectile, respectively.

#### 4.1.2. Simulation Analysis

The simulation calculation of the standard projectile and ZTA ceramic composite projectile penetrating a 15 mm armored steel target was carried out when the initial velocity of the projectile was 1000 m/s. The damage effects of the two projectiles on the armored steel target are shown in Figure 11. It is noted that while both armor-piercing projectiles can penetrate 15 mmRHA, the ZTA ceramic composite projectile had a smaller steel core mass loss and larger penetration aperture than the standard projectile.

See Table 5 for the simulation data of armor-piercing projectiles with two structures penetrating a 15 mm armored steel target. From this table, the core residual mass of the standard projectile and ZTA ceramic composite projectile were 19.5 g and 25.6 g, respectively, with their respective mass-loss rate being 51.61% and 28.49%, and the storage velocity after penetration being 388.8 m/s and 493.9 m/s. The residual velocity of the ZTA composite projectile was 27.03% higher than that of the standard projectile. Figure 12 and Figure 13 show the mass-time and velocity/acceleration-time curves of the cores with two structures, respectively. When penetrating the Al_2_O_3_/RHA composite armor in these two curves, the standard bullet core suffered severe mass loss and a slower residual speed [29]. In the penetration process, the ZTA composite armor-piercing projectile had noted advantages over the standard projectile in the residual mass and storage speed. This is because, in the process of penetrating armor, the ZTA ceramic projectile warhead acts on the target plate first, which effectively protects the core, and hence its loss of core mass and speed is significantly less than that of the standard projectile. The standard projectile core directly acts on the ceramic plate. The upsetting bullet tip results in an increase in the penetration resistance, and the bullet core is seriously eroded by the target plate. In the comparison of the two acceleration-time curves, it is shown that the acceleration of the two armor-piercing projectiles basically reaches the same peak in the penetration process, but the ZTA composite armor-piercing projectile takes a shorter time. The ZTA ceramic warhead effectively protects the integrity of the core in the early stage of high-speed penetration.

From the tests and simulation results, it is noted that the ZTA warhead effectively protected the metal core when penetrating 15 mm armor steel, so that the ZTA ceramic composite armor-piercing projectile significantly exceeds the standard armor-piercing projectile in hole diameter, residual mass, and residual velocity.

### 4.2. Penetration of Al_2_O_3_/RHA Composite Armor

#### 4.2.1. Test Results

In order to analyze the penetration ability of the 14.5 mm ZTA ceramic composite armor-piercing projectile and the standard armor-piercing projectile to Al_2_O_3_/RHA composite armor, the penetration power test was carried out at the speed of 1000 ± 15 m/s. The test was repeated three times, and the average result was used as the criterion for evaluating the projectile power. The test data of two kinds of projectiles penetrating Al_2_O_3_/RHA composite armor are shown in Table 6, and the typical damage of ceramic/armor steel composite target armor steel is shown in Figure 14.

When penetrating the ceramic composite armor using the standard projectile, the armor-piercing projectile core was broken and splashed, and the ceramic panel was broken into ceramic cones [30,31] under the high-speed impact, in which the ceramic block was further shattered and squeezed by the remaining projectile body and continuously gained speed in the direction of the impact. Finally, the remaining projectile and ceramic cone fragments acted on the armor steel plate together, leaving an impact trace (radius of about 35 mm) surrounding the impact point. The crater was 2 mm deep, and there was no damage trace on the back of the armor steel, which proves the excellent anti-penetration properties of Al_2_O_3_/RHA composite armor to the metal projectile. When the ZTA ceramic composite projectile penetrated Al_2_O_3_/RHA composite armor at the same speed, the ceramic layer was broken into ceramic blocks, and the projectile penetrated completely the backplate of the composite target, causing a filling. With ultra-high hardness, high abrasion, and impact toughness, the ceramic warhead destroys the ceramic layer ideally when the projectile penetrates the ceramic composite target at high speed, thus offering a channel for the core to penetrate the backplate of the composite target. In this way, the core is integrated, promising a higher penetration ability of the composite projectile.

The typical ceramic target fragments and projectiles recovered are shown in Figure 15. It is noted that the standard projectile obviously damaged the ceramic target, which was crushed into smaller particles, and its bullet core was disintegrated into small metal blocks. A small residual mass of the copper shell that covers the bullet core indicates a huge energy loss of the standard projectile. A serious projectile disintegration is caused when the standard projectile penetrates the ceramic panel of the composite armor. In contrast, the ZTA composite bullet did less damage to the ceramic panel of the composite armor with relatively uniform ceramic particles. Although the ZTA warhead was broken, more main parts of the core were kept. Furthermore, the energy was better preserved for the subsequent penetration of armor steel. From the main part of the core, 11.3 g and 18.4 g, on average, were recovered in the standard armor-piercing projectile and ZTA composite armor-piercing projectile, representing 28.04% and 51.40% of the initial core mass. The average mass loss of the ZTA composite projectile was 66.67% less than that of the standard projectile. The standard projectile failed to penetrate the armor steel backplate, and the energy was exhausted. However, the ZTA ceramic armor-piercing projectile penetrated the armor steel backplate with an average residual velocity of 241.1 m/s and residual kinetic energy of 534.79 J. The ZTA composite projectile outshines the standard projectile in penetration power.

#### 4.2.2. Simulation Analysis

A simulation analysis was carried out when the Al_2_O_3_/RHA was penetrated by the standard armor-piercing projectile and ZTA ceramic composite projectile with an initial velocity of 1000 m/s. The penetration process of the ZTA ceramic composite armor-piercing projectile into Al_2_O_3_/RHA composite armor is shown in Figure 16.

When t = 20 μs, the ZTA ceramic composite bullet penetrated the front glass fiber layer and acted on the ceramic panel. Because the ZTA ceramic at the warhead was harder than Al_2_O_3_ ceramic, the ceramic panel showed brittle fracture and broke into blocks under the action of high-speed impact, forming a ceramic cone [32,33]. With deeper penetration, the ZTA warhead was crushed and acted on the target plate continuously when the compressive wave strength exceeded the failure limit of ZTA ceramics; when t = 50 μs, the ceramic block continued to crush under the interface impact pressure, and the crack propagation extended to the edge of the target plate. The metal core was intact under the protection of the ZTA warhead, began to penetrate the armored steel target, and then started to suffer mass loss. When t = 90 μs, under the action of a compression wave, there were pits on the front, and large deformations on the back of the steel target plate, and the erosion of the target plate to the bullet core was further intensified. Then, shear cracks began to appear on the contact surface of the projectile target and extend to the back of the target plate; when t = 110 μs, the shear crack penetrated, and fillings were formed. The metal core penetrated the target plate.

The damage effects of two armor-piercing projectiles on Al_2_O_3_/RHA composite armor are shown in Figure 17. (See Table 7 for the simulation data of armor-piercing projectiles with two structures penetrating composite targets.) After the penetration, the core residual masses of the standard projectile and the ZTA ceramic composite projectile were 10.8 g and 17.4 g, with a core mass loss of 73.2% and 51.4%, respectively. The residual mass of the core of the ZTA ceramic composite armor-piercing projectile was 61.1% higher than that of the standard one. When penetrating the ceramic layer, the ceramic composite projectile had less mass loss than the standard armor-piercing projectile. The ceramic warhead with high hardness destroyed the ceramic layer of the composite target under the high-speed impact. Less action time is needed between the core and the ceramic layer, and the abrasion of the core is reduced. The simulation results are consistent with the experimental results. 

Figure 18 and Figure 19 show the mass-time and velocity/acceleration-time curves of the two projectiles, respectively, according to which, the standard armor-piercing projectile had a faster mass loss and velocity drop compared with the ZTA ceramic composite armor-piercing bullet. The overload of the ZTA ceramic composite armor-piercing bullet core was significantly below that of the standard armor-piercing bullet core. This is because the ceramic warhead acts on the target plate at the moment when the ZTA ceramic composite armor-piercing bullet contacts the target plate, which effectively protects the metal bullet core.

From the two groups of tests and simulation results, it can be concluded that the high-strength ZTA ceramic warhead protects the metal core to a certain extent so that ZTA ceramic composite armor-piercing projectile has less core mass loss and superior penetration power compared to the standard one.

## 5. Penetration Power Calculation

The theoretical research on the penetration power of an armor-piercing projectile into a ceramic composite target helps predict the penetration depth and mass loss of the projectile under different initial velocities. The Alekseevskii–Tate (A-T) model [34,35,36] has been widely used in the terminal effect analysis of the high-speed impact of the armor-piercing projectile. The projectile strength (*Y_p_*) and target strength (*R_t_*) are introduced into the Bernoulli equation to establish the semi-fluid hydrodynamic model:(1)$$\frac{1}{2}\rho_{p}{\left(v_{p}-u\right)}^{2}+Y_{P}=\frac{1}{2}\rho_{t}u^{2}+R_{t}$$

In this equation, *v_p_* is the instantaneous penetration velocity of the tail of the projectile; *u,* the instantaneous penetration velocity of the head of the projectile (the contact surface between the projectile and the target); *ρ_p_*, *ρ_t_*, the density of projectile and target materials, respectively.

Supposing that the contact interface between the projectile and the target is in a fluid state and the projectile body is eroded in the process of high-speed penetration, the velocity decline equation of the projectile body representing the length *L_p_* and the motion equation of motion of the projectile body mass loss rate are as follows:(2)$$\frac{dv_{p}}{dt}=-\frac{Y_{P}}{\rho_{p}L_{p}}$$
(3)$$\frac{dL_{p}}{dt}=-(v_{p}-u)$$

The equation on the penetration depth of the projectile is:(4)$$\frac{dP}{dt}=u$$

The penetration depth of a projectile hitting a semi-infinite target at different speeds can be obtained by solving the equations composed of Equations (1)–(4). In order to simplify the calculation, nondimensionalize the equations. Let $\bar{t}=\frac{t}{t_{0}}$, $\bar{L}=\frac{L_{p0}}{L_{p}}$, ${\bar{Y}}_{p}=\frac{Y_{P}}{\rho_{p}v_{p0}^{2}}$, ${\bar{R}}_{t}=\frac{R_{t}}{\rho_{p}v_{p0}^{2}}$, $t_{0}=\frac{L_{p0}}{v_{p0}}$, ${\bar{v}}_{p}=\frac{v_{p}}{v_{p0}}$, $\bar{P}=\frac{P}{L_{p0}}$, $\xi =\frac{\rho_{t}}{\rho_{p}}$, then we can get:(5)$$\left\{\begin{array}{c} \frac{d{\bar{v}}_{p}}{d\bar{t}}=-\frac{{\bar{Y}}_{P}}{{\bar{L}}_{p}}\\ \frac{d{\bar{L}}_{p}}{d\bar{t}}=-({\bar{v}}_{p}-\bar{u})\\ \frac{d\bar{P}}{d\bar{t}}=\bar{u}\\ \bar{u}=\frac{{\bar{v}}_{p}}{1-\xi }(1-\sqrt{\xi +2(1-\xi )({\bar{R}}_{t}-{\bar{Y}}_{p}){\bar{v}}_{p}^{-2}} \end{array}\right.$$

In this equation, *L_p_*_0_ refers to the initial length of the projectile and *v_p_*_0_ to its initial velocity. The initial condition for solving the equations is:$${\bar{t}}_{0}=0,\;{\bar{L}}_{p0}=1,\;{\bar{v}}_{p0}=1,\;{\bar{P}}_{0}=0$$

According to the first, second, and fourth equations in Equation (5), the relationship between dimensionless velocity ${\bar{v}}_{p}$ and ${\bar{L}}_{p}$ can be obtained as follows:(6)$$\ln{\bar{L}}_{p}=\frac{1}{{\bar{Y}}_{p}}\int_{1}^{{\bar{v}}_{p}}({\bar{v}}_{p}-\bar{u}({\bar{v}}_{p}))d{\bar{v}}_{p}$$

From Equation (6), the relationship between the residual mass ${\bar{M}}_{p}$ of the dimensionless projectile and the velocity ${\bar{v}}_{p}$ of the projectile body can be figured out. Among them, $b=2(1-\xi )({\bar{R}}_{t}-{\bar{Y}}_{p})$.
(7)$$\begin{array}{l} {\bar{M}}_{p}=\pi \cdot {\left(d/2\right)}^{2}\cdot \rho_{p}\cdot \exp\left\{\frac{\xi \cdot {\bar{v}}^{2}{}_{p}}{2(1-\xi )}-\frac{{\bar{v}}_{p}(\sqrt{\xi {\bar{v}}^{2}{}_{p}+b}\cdot \sqrt{\xi })+bln({\bar{v}}_{p}\sqrt{\xi }+\sqrt{\xi {\bar{v}}^{2}{}_{p}+b})}{2\sqrt{\xi }}\right\}\cdot \\ \exp\left\{-\frac{\xi }{2(1-\xi )}+\frac{\sqrt{\xi +b}\cdot \sqrt{\xi }+b\ln(\sqrt{\xi +b}+\sqrt{\xi })}{2\sqrt{\xi }}\right\} \end{array}$$

From the first, third, and fourth equations in the Equation (5), the relationship between the dimensionless penetration depth $\bar{H}$ and projectile velocity ${\bar{v}}_{p}$ can be obtained as follows: (8)$$\bar{H}=\int_{0}^{\bar{t}}\bar{u}d\bar{t}=-\frac{1}{{\bar{Y}}_{p}}\int_{1}^{{\bar{v}}_{p}}{\bar{L}}_{p}\cdot \bar{u}d{\bar{v}}_{p}$$

Taking the traditional quasi-static cavity expansion theory as the basis and considering the loss factor and loss state, we divided the response after the ceramic layer impacted into cavity area, crushing area, radial crack area, elastic area, and unresponsive area. The resistance calculation formula [8] of the ceramic/metal composite target is deduced as follows:(9)$$R_{t}=Y{\left[\frac{E_{0}/\left(3Y\right)}{(1+\nu )\sqrt{\sigma_{f}/2Y}+\frac{1}{4}{\left(\sqrt{Y/2\sigma_{f}}\right)}^{3}-\frac{1}{4}{\left(\sqrt{2\sigma_{f}/Y}\right)}^{2}}\right]}^{2\alpha /3}$$

In this equation, *α* = 6*λ*/(3 + 4*λ*), *λ* represents the shear coefficient of ceramic material; *ν* refers to its Poisson’s ratio; *E*_0_ is its initial elastic modulus; *σ_f_* its tensile strength; and *Y* its compressive strength. MATLAB was utilized to determine the penetration depth and core residual mass of the standard projectile and ZTA ceramic composite projectile penetrating the Al_2_O_3_/RHA composite target. The material parameters of the target plate refer to those in literature [8]. The elastic modulus of the ceramic material was 268.9 GPa; the density 3.625 g/cm^3^; the Poisson’s ratio 0.228; the compressive strength 2.03 GPa; the tensile strength 0.262 GPa; the shear coefficient 0.273; and the calculated penetration resistance 4.548. The strength parameter of the standard projectile *Y_p_* = 1.6 Gpa, and the strength parameter of the composite projectile *Y_p_* = 2 GPa. Figure 20 illustrates the variation curve of the velocity between the penetration depth of the two kinds of composite projectiles to the composite target and the core residual mass.

As shown in Figure 20, a growing initial velocity means a more considerable penetration depth and a lower mass of the armor-piercing projectile. Under the same initial conditions, the penetration depth and core residual mass of the ZTA ceramic composite projectile exceeded those of the standard projectile. When the initial velocity was 1000 m/s, the penetration depth of the ZTA ceramic composite projectile was 22.47 mm, and the residual mass was 13.89 g. The penetration depth of the standard projectile was 18.84 mm, and the residual mass was 10.93 g. The penetration depth of the ZTA ceramic composite projectile increased by 19.27%, and the core mass loss decreased by 34.05%, which is basically consistent with the test results.

## 6. Conclusions

The ZTA ceramic projectile prepared in this study has ideal impact toughness, and ZTA ceramic parts are applied to armor-piercing warheads and assembled with standard bullet cores to prepare ZTA ceramic composite bullets. Through tests, numerical simulations, and theoretical analyses, we studied the penetration of 14.5 mm armor-piercing standard projectile and ZTA ceramic composite armor-piercing projectile into a 15 mm armor steel target and Al_2_O_3_/RHA composite armor at a speed of 1000 ± 15 m/s. The results show that: (1) Compared with the standard projectile penetrating RHA, the average area of entry into the RHA by the ZTA ceramic composite ammunition increased by 27.59%, and the average area of exit from the hole increased by 42.93%. (2) Under the same initial condition, the standard projectile fails to penetrate the ceramic composite armor. As for the damage caused to the Al_2_O_3_/RHA composite armor, the ZTA ceramic composite armor-piercing projectile penetrates the ceramic composite armor with an average residual velocity of 241.1 m/s, with an average mass loss reduced by 66.67% over the standard projectile. The ZTA ceramic composite armor-piercing projectile is superior to the standard one in the penetration of ceramic composite armor. ZTA ceramic material, as an armor-piercing warhead, if designed reasonably to match the traditional armor-piercing core, will have good prospects for application in the field of anti-ceramic composite armor.

## Figures and Tables

**Figure 1 materials-15-02909-f001:**
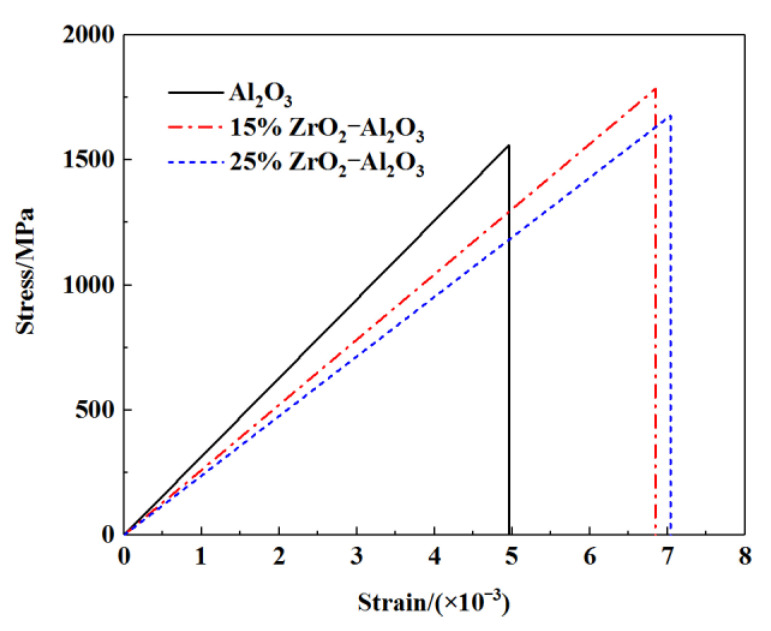
The specific change trend of the mechanical properties of ZTA with ZrO_2_ content.

**Figure 2 materials-15-02909-f002:**
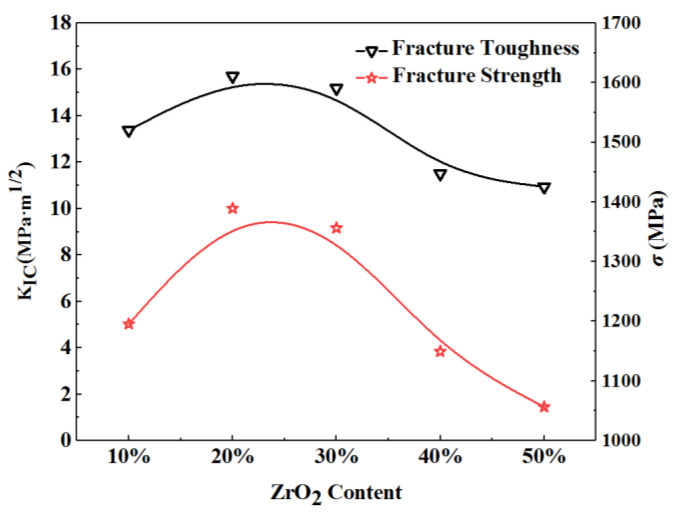
The influence of ZrO_2_ content on ZTA fracture.

**Figure 3 materials-15-02909-f003:**
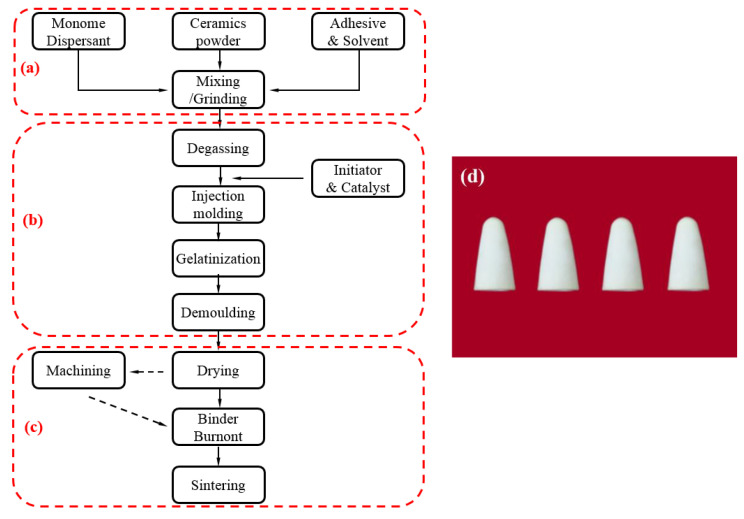
Flow chart of ZTA ceramic projectile forming process. (**a**) Powder mixing process; (**b**) molding process; (**c**) sintering process; (**d**) prepared ZTA bullet tip samples.

**Figure 4 materials-15-02909-f004:**
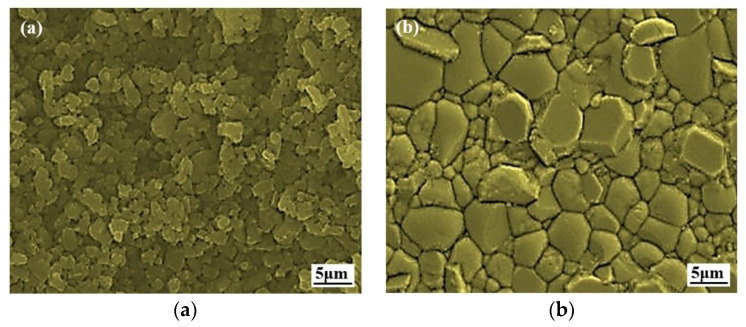
Preparation of green body and micro-structure of ZTA ceramics: (**a**) ZTA green body; (**b**) ZTA ceramic polished surface.

**Figure 5 materials-15-02909-f005:**
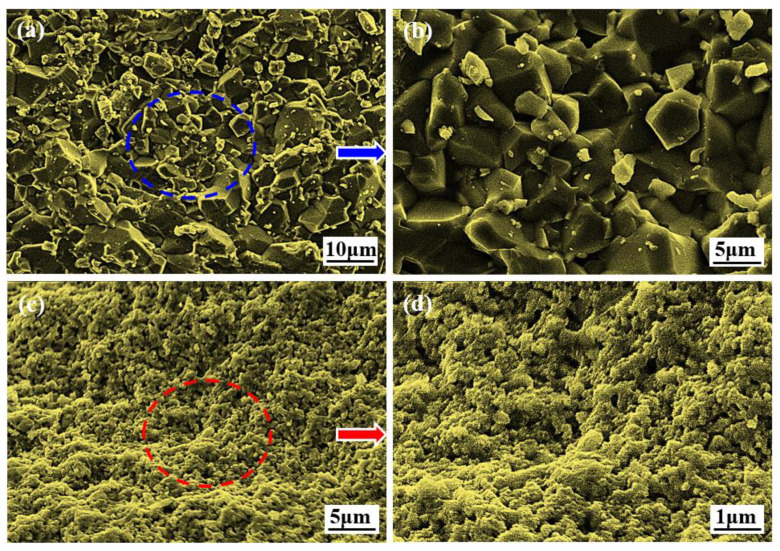
The micro appearance of fracture of Al_2_O_3_ ceramics (**a**,**b**) and ZTA ceramics (**c**,**d**) after high-speed impact.

**Figure 6 materials-15-02909-f006:**
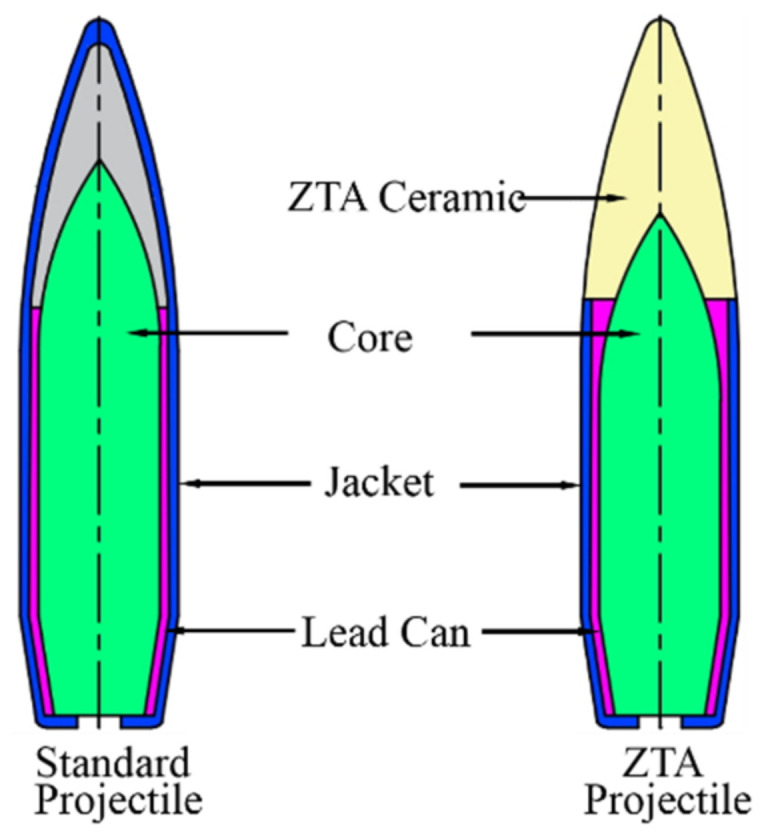
Two structures of projectiles.

**Figure 7 materials-15-02909-f007:**
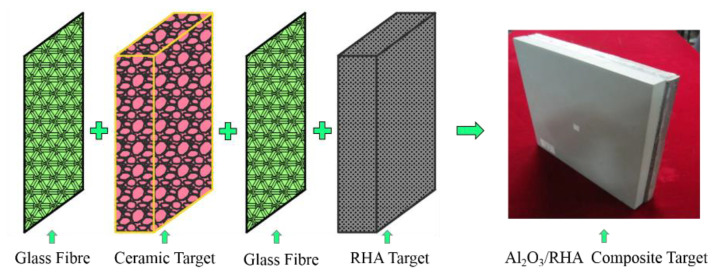
Al_2_O_3_/RHA Composite armor structure diagram.

**Figure 8 materials-15-02909-f008:**
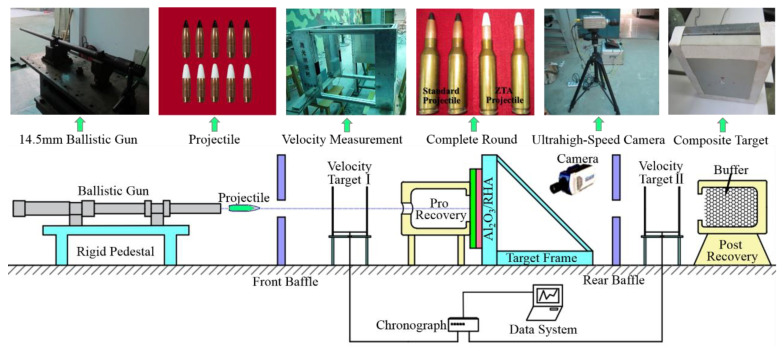
Diagram of the experiment setup.

**Figure 9 materials-15-02909-f009:**
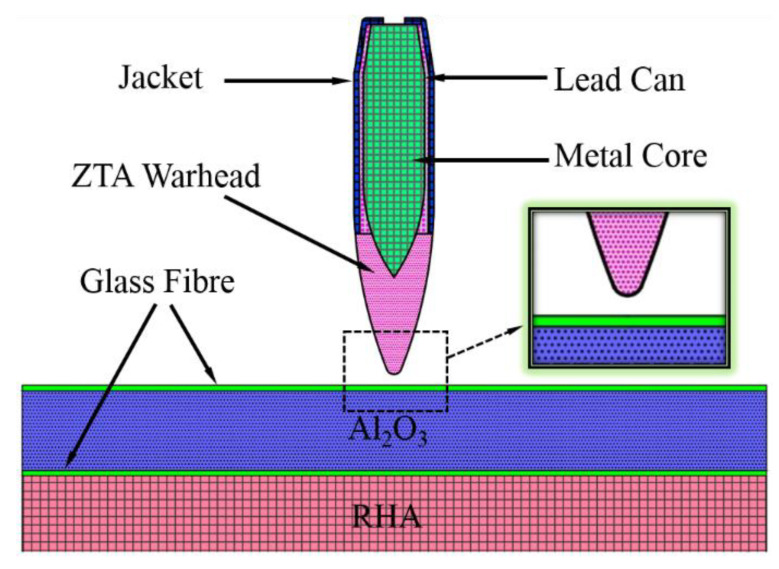
The projectile target simulation model.

**Figure 10 materials-15-02909-f010:**
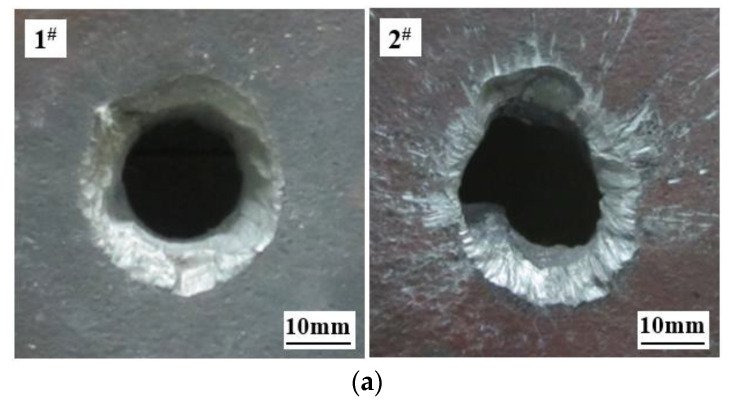
Comparison of typical damage effects of armor steel target: (**a**) Front of target plate; (**b**) back of target plate. 1^#^ denotes the damage effect of standard projectile; 2^#^ denotes the damage effect of ZTA composite projectile.

**Figure 11 materials-15-02909-f011:**
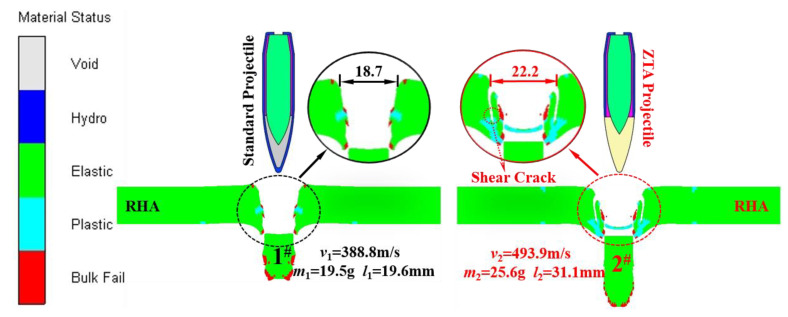
The damage effects of the two projectiles on the armored steel target. 1^#^ denotes the remaining core of standard projectile; 2^#^ denotes the remaining core of ZTA composite projectile.

**Figure 12 materials-15-02909-f012:**
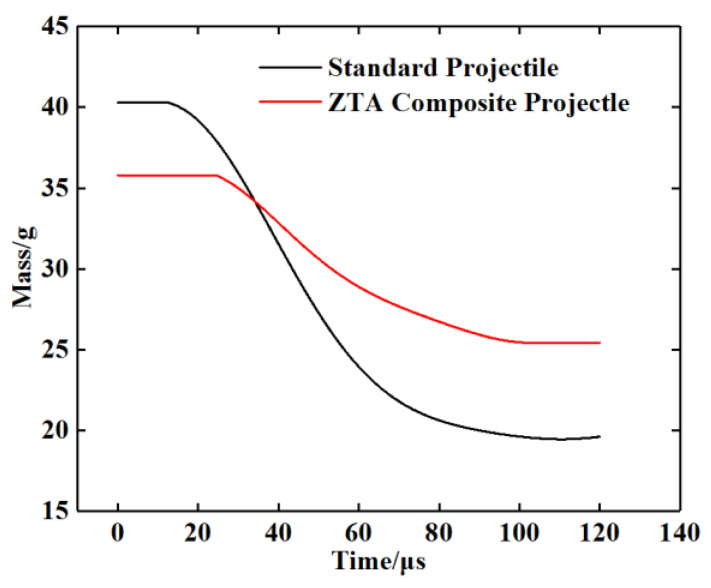
Projectile core mass-time curve.

**Figure 13 materials-15-02909-f013:**
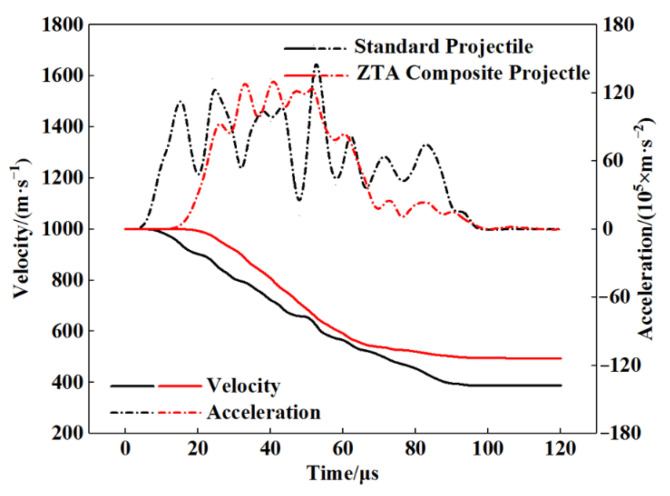
Projectile core velocity/acceleration-time curve.

**Figure 14 materials-15-02909-f014:**
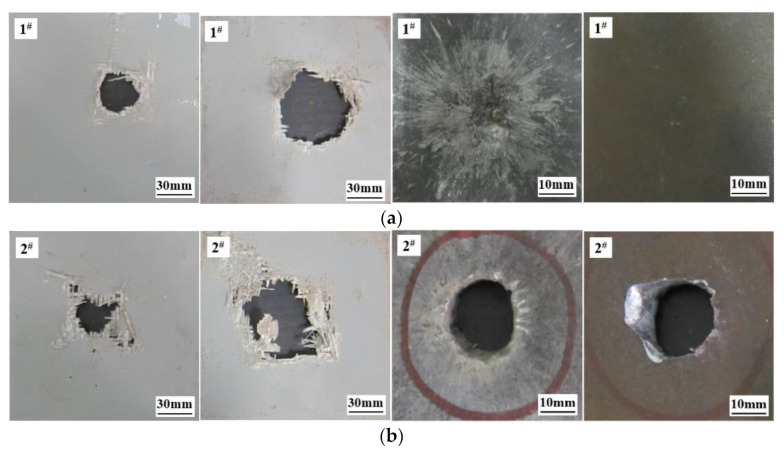
Typical damage of Al_2_O_3_/RHA composite armor. (**a**) Typical damage effect of standard projectile on Al_2_O_3_/RHA composite armor. (**b**) Typical damage effect of ZTA ceramic composite projectile on Al_2_O_3_/RHA composite armor. 1^#^ denotes the damage effect of standard projectile; 2^#^ denotes the damage effect of ZTA composite projectile.

**Figure 15 materials-15-02909-f015:**
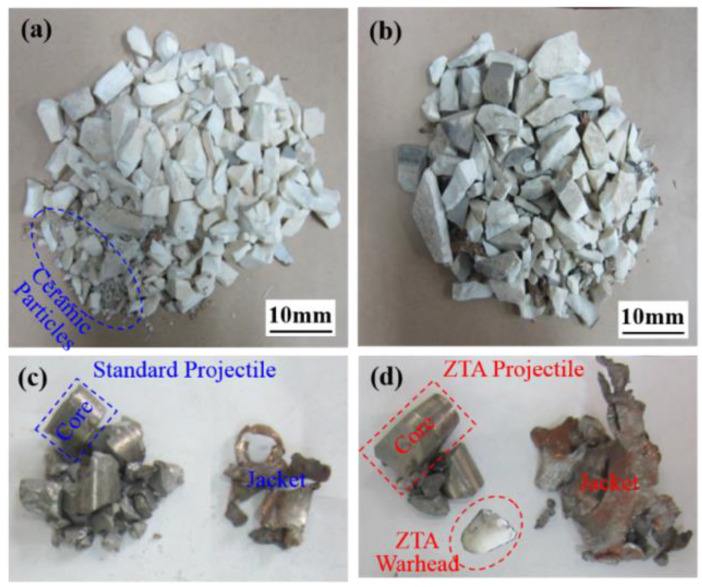
Recovered typical ceramic target fragments (**a**,**b**) and projectiles (**c**,**d**).

**Figure 16 materials-15-02909-f016:**
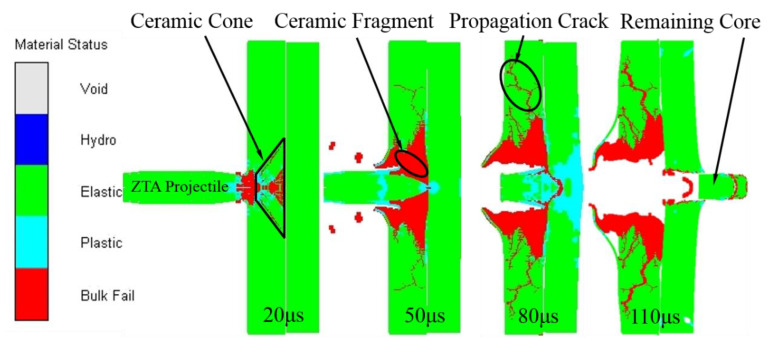
Penetration process of ZTA ceramic composite bullet into Al_2_O_3_/RHA composite armor.

**Figure 17 materials-15-02909-f017:**
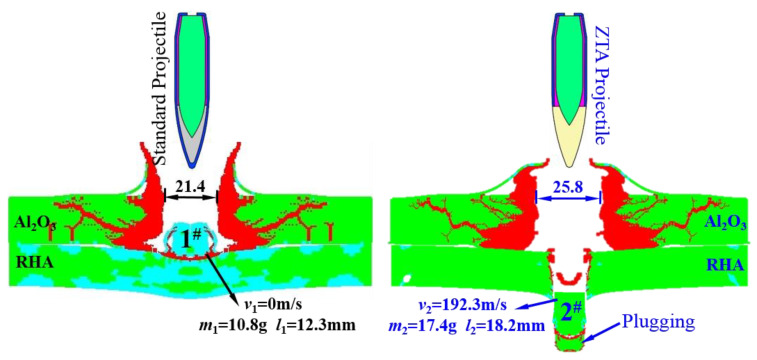
Destruction effect of two armor-piercing projectiles on Al_2_O_3_/RHA composite armor. 1^#^ denotes the remaining core of standard projectile; 2^#^ denotes the remaining core of ZTA composite projectile.

**Figure 18 materials-15-02909-f018:**
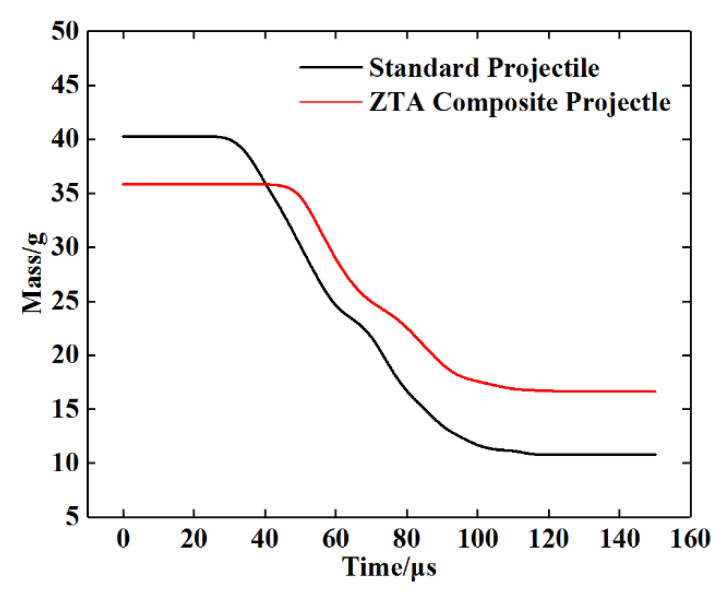
Projectile core mass-time curve.

**Figure 19 materials-15-02909-f019:**
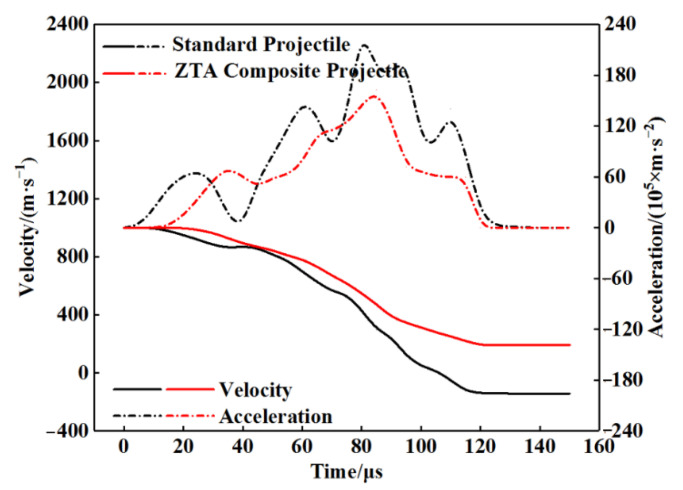
Projectile core velocity/acceleration-time curve.

**Figure 20 materials-15-02909-f020:**
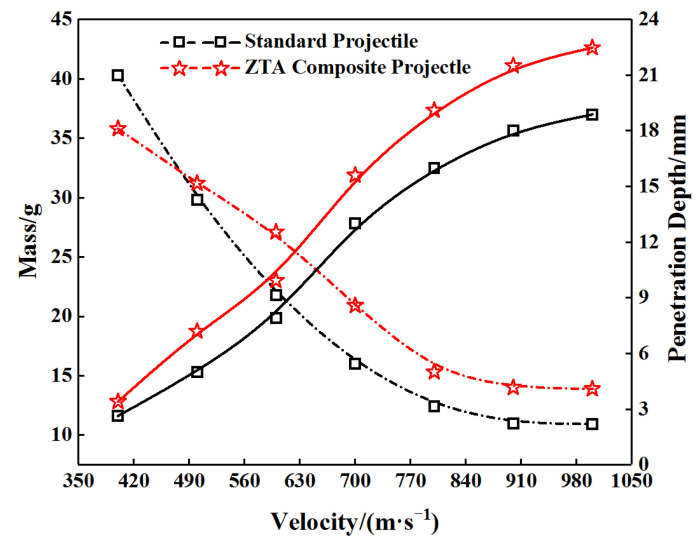
Variation curve of the velocity between penetration depth of the two kinds of composite projectiles to the composite target and the core residual mass.

**Table 1 materials-15-02909-t001:** Physical and mechanical properties of ZTA ceramics.

Material	Relative Density	Flexural Strength (MPa)	Fracture Toughness (MPa·m^1/2^)	Vickers Hardness (GPa)
ZTA Ceramics	98.5%	1026	9.65	17.39

**Table 2 materials-15-02909-t002:** Parameters of projectile structures.

No.	Projectile Type	Projectile			Metal Core		
		Diameter /mm	Length /mm	Mass /g	Diameter /mm	Length /mm	Mass /g
1^#^	Standard Projectile	14.9	66.7	63.9	12.4	52.4	40.3
2^#^	ZTA Composite Projectile	14.9	66.7	63.9	12.4	47.6	35.8

Note: 1^#^ denotes the parameters of standard projectile; 2^#^ denotes the parameters of ZTA composite projectile.

**Table 3 materials-15-02909-t003:** Ceramic material parameters.

Materials	*ρ*/(g·cm^−3^)	G/GPa	HEL/GPa	A	B	C	M	N
Al_2_O_3_	3.93	90.1	4.31	0.90	0.31	0.007	0.35	0.60
ZTA	5.90	152	6.57	0.93	0.72	0.007	0.38	0.64

Note: G is the shear modulus; ***ρ*** is density; A is the intact strength parameter; B is the fracture strength parameter; M is the strain strength parameter; C is the strain rate strength parameter; N is the pressure fracture strength parameter; HEL is the elastic limit.

**Table 4 materials-15-02909-t004:** Test data of projectiles with two structures penetrating 15 mm armored steel target.

No.	Projectile Structure	Core Initial Mass	Initial Velocity (m/s)	Core Residual Mass	Leaving Velocity (m/s)	Damage of Armored Steel Target	
						Entrance (mm)	Exit (mm)
1^#^-a	Standard projectile	40.3 g	1012.4	20.6 g	450.9	Ø 19.2	Ø 15.6
1^#^-b			1003.2	19.8 g	423.6	Ø 19.3	Ø 15.8
1^#^-c			1011.8	21.1 g	427.8	Ø 18.2	Ø 16.9
2^#^-a	ZTA Composite Projectile	35.8 g	1011.7	29.8 g	514.4	22.9 × 15.7	17.8 × 15.7
2^#^-b			1005.9	21.3 g	522.3	21.9 × 16.3	17.7 × 15.4
2^#^-c			1009.1	26.6 g	513.4	23.6 × 15.1	19.4 × 16.6

Note: 1^#^ denotes the test data of standard projectile; 2^#^ denotes the test data of ZTA composite projectile.

**Table 5 materials-15-02909-t005:** Simulation data of two kinds of projectiles penetrating 15 mm armored steel target.

No.	Projectile Structure	Core Initial Mass	Initial Velocity (m/s)	Core Residual Mass	Leaving Velocity (m/s)	Damage of Armored Steel Target	
						Hole Entry (mm)	Hole Exit (mm)
1^#^	Standard Projectile	40.3 g	1000	19.5 g	388.8	Ø 18.7	Ø 14.9
2^#^	ZTA Composite Projectile	35.8 g	1000	25.6 g	493.9	Ø 22.2	Ø 15.6

Note: 1^#^ denotes the simulation data of standard projectile; 2^#^ denotes the simulation data of ZTA composite projectile.

**Table 6 materials-15-02909-t006:** The test data of two kinds of projectiles penetrating Al_2_O_3_/RHA composite armor.

No.	Projectile Structure	Core Initial Mass	Initial Velocity (m/s)	Core Residual Mass	Leaving Velocity (m/s)	Damage Effect of Al_2_O_3_/RHA Composite Target
1^#^-a	Standard Projectile	40.3 g	998.2	11.2 g	0	The projectile penetrated the glass fiber and Al_2_O_3_ ceramic layer but did not penetrate the armored steel plate. The average tear range of glass fiber was Ø 33.2 mm for the front surface and Ø 64.7 mm for the back surface. The Al_2_O_3_ ceramic panel was broken, and there were many crushed ceramic blocks. The front armor steel had Ø 35 mm impact marks; the average size of the crater was 5.6 mm × 4.2 mm with a depth of 2.2 mm; there was no damage to the back.
1^#^-b			1004.4	10.8 g	0	
1^#^-c			1009.8	11.9 g	0	
2^#^-a	ZTA Composite Projectile	35.8 g	1001.7	18.4 g	234.0	The projectile completely penetrated the Al_2_O_3_/RHA composite armor. The average tear range of glass fiber was 37.9 mm × 25.3 mm on the front surface, and 90.7 mm × 63.2 mm on the back surface. The Al_2_O_3_ ceramic panel was broken; the size of fragments was relatively uniform. Fillings were formed when the projectile penetrated the armor steel; the average sizes of the front and back bullet holes were 21.5 mm × 19.2 mm and 22.3 mm × 16.9 mm.
2^#^-b			996.9	17.6 g	231.4	
2^#^-c			1007.4	19.2 g	257.9	

Note: 1^#^ denotes the simulation data of standard projectile; 2^#^ denotes the simulation data of ZTA composite projectile.

**Table 7 materials-15-02909-t007:** Simulation data of penetration of projectile with two structures into composite target.

No.	Projectile Structure	Core Initial Mass	Initial Velocity (m/s)	Core Residual Mass	Leaving Velocity (m/s)	Damage Effect of Al_2_O_3_/RHA Composite Target
1^#^	Standard Projectile	40.3 g	1000	10.8 g	0	The Al_2_O_3_ ceramic panel was broken; the damage range was Ø 119.6 mm. The projectile failed to penetrate the back target of the armored steel; the penetration depth was 1.3 mm.
2^#^	ZTA Composite Projectile	35.8 g	1000	17.4 g	192.3	The Al_2_O_3_ ceramic panel was broken with Ø 109.7 mm of the damage range. The projectile penetrated the back target of the 15 mm armored steel and formed a filling.

Note: 1^#^ denotes the simulation data of standard projectile; 2^#^ denotes the simulation data of ZTA composite projectile.

## Data Availability

Some or all data, models, or code that support the findings of this study are available from the corresponding author upon reasonable request (Data in Table 4 and Table 6).

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
