# Peer review of "Study on the Penetration Power of ZrO2 Toughened Al2O3 Ceramic Composite Projectile into Ceramic Composite Armor"

_materials, 2022, doi:10.3390/ma15082909_

Round 1
Reviewer 1 Report
Great job,
- However, I ask you to expand the introduction and add alternative methods for obtaining ceramic materials with their further sintering, etc. until the finished product is obtained.
- Give graphs of phase and chemical analysis, as well as additional explanations, i.e. the effect of the initial concentration of reagents on the formation of certain phases.
In my opinion, it is necessary to expand the introduction by adding alternative methods for obtaining powders of this type.
I would also like to see an analysis of the change in the phase composition and its description. For example, alumina is stabilized when a more significant amount of zirconium oxide is introduced.
I also wanted to see not only morphology but also individual mapping, the distribution of chemical components over the surface, and their uniform distribution.
Author Response
Thank you very much for your patience in reviewing this paper and the helpful comments provided. Based on these constructive comments, we have made the following revisions to the original manuscript. Below is our point-by-point response to the comments.The specific modifications are shown in the attachment.

Reviewer 2 Report
Information for Authors
The results obtained in the reviewed article on the basis of the experiment are a novelty.
To supplement some lack of information’s I formulate a few remarks:
- Line 75: What do You mean “field tanks”? Probably You would like write “battle tanks” as example of the armament equipment.
- Line 115: The Fig 1 and 2 haven’t enough resolution. Please correct it.
- Line 209: There is no description of the all adopted material constants for the simulated test system, no failure criteria etc. Please supplement data used in simulations for all materials (RHA, metal core, jacket etc.)
In connection with the above, I propose to extend the bibliography with a two new similar articles containing also the numerical simulations, especially in terms of the description of the input data for the simulation.
- Zochowski, P., Pyka, D., et all., Comparison of Numerical Simulation Techniques of Ballistic Ceramics under Projectile Impact Conditions, 2022, 15, 18. https://doi.org/10.3390/ma15010018
- Pyka, D., et all., Assessment of the Impact Resistance of a Composite Material with EN AW-7075 Matrix Reinforced with α-Al2O3 Particles Using a 7.62 x 39 mm Projectile, 2020, 13, 769. https:// doi:10.3390/ma13030769.
- Line 280: The Fig 12 and 13 haven’t enough resolution. Please correct it. Moreover “Atandard” correct to “Standard”.
- Line 337: There is no description of the Al2O3 and glass fibre material data used in simulation.
- Line 382: The Fig 18 and 19 haven’t enough resolution. Please correct it. Moreover “Atandard” correct to “Standard”.
- Line 437: “Atandard” correct to “Standard”.
Author Response

(The authors gave the same response as above.)

Reviewer 3 Report
The manuscript needs some serious English correction. The manuscript may be accepted after language correction is made. I dont have any mechanism to check plagiarism. To the extent I can explore I feel that the manuscript consists of original contents.
Author Response

(The authors gave the same response as above.)

Reviewer 4 Report
Authors studied the Penetration Power of ZTA Ceramic Composite Projectile into Al2O3/RHA Composite Armor and reported the ZTA ceramic composite bullet has a better performance than standard bullet in penetrating RHA and Al2O3/RHA composite armor. This paper can be accepted after minor revision.
- Please change the title of the paper without abbreviation.
- Avoid abbreviation in keywords
- Check the language of the paper thoroughly
- How the property values are provided in section 2.1?
- The font size for text available in line graph is looking tiny. Please check and improve
- 3 Flow chart of ZTA ceramic projectile forming process – provide the sample photo images also.
- Figure 4- provide the scale bar clearly
- How the relative density was found? Provide the density values clearly
- MPa or Mpa – check thoroughly
- Figure 5- provide the scale bar clearly
- Table 6 The test data of two kinds of projectiles penetrating Al2O3 – provide the materials name properly. Check this throughout the manuscript.
- In discussion citing of others research is missing. Please cite the references suitably and improve the discussion.
- Revise the conclusion concisely.
Author Response

(The authors gave the same response as above.)
